# Associations of metabolic syndrome and metabolically unhealthy obesity with cancer mortality: The Japan Multi-Institutional Collaborative Cohort (J-MICC) study

Tien Van Nguyen[1], Kokichi Arisawa[1]*, Sakurako Katsuura-Kamano[1], Masashi Ishizu[1], Mako Nagayoshi[2], Rieko Okada[2], Asahi Hishida[2], Takashi Tamura[2], Megumi Hara[3], Keitaro Tanaka[3], Daisaku Nishimoto[4], Keiichi Shibuya[4], Teruhide Koyama[5], Isao Watanabe[5], Sadao Suzuki[6], Takeshi Nishiyama[6], Kiyonori Kuriki[7], Yasuyuki Nakamura[8], Yoshino Saito[9], Hiroaki Ikezaki[10], Jun Otonari[11], Yuriko N. Koyanagi[12], Keitaro Matsuo[12], Haruo Mikami[13], Miho Kusakabe[13], Kenji Takeuchi[2], Kenji Wakai[2]

1 Department of Preventive Medicine, Tokushima University Graduate School of Biomedical Sciences, Tokushima, Japan, 2 Department of Preventive Medicine, Nagoya University Graduate School of Medicine, Nagoya, Japan, 3 Department of Preventive Medicine, Faculty of Medicine, Saga University, Saga, Japan, 4 Department of International Island and Community Medicine, Kagoshima University Graduate School of Medical and Dental Sciences, Kagoshima, Japan, 5 Department of Epidemiology for Community Health and Medicine, Kyoto Prefectural University of Medicine, Kyoto, Japan, 6 Department of Public Health, Nagoya City University Graduate School of Medical Sciences, Nagoya, Japan, 7 Division of Nutritional Sciences, Laboratory of Public Health, School of Food and Nutritional Sciences, University of Shizuoka, Shizuoka, Japan, 8 Department of Public Health, Shiga University of Medical Science, Otsu, Shiga, Japan, 9 Department of Nursing, Faculty of Healthcare Science, Aino University, Osaka, Japan, 10 Department of Comprehensive General Internal Medicine, Faculty of Medical Sciences, Kyushu University Graduate School of Medicine, Fukuoka, Japan, 11 Department of Psychosomatic Medicine, Graduate School of Medical Sciences, Kyushu University, Fukuoka, Japan, 12 Division of Cancer Epidemiology and Prevention, Aichi Cancer Center Research Institute, Nagoya, Japan, 13 Cancer Prevention Center, Chiba Cancer Center Research Institute, Chiba, Japan

* karisawa@tokushima-u.ac.jp

## Abstract

### Purpose

The association between metabolic syndrome (MetS) and the risk of death from cancer is still a controversial issue. The purpose of this study was to examine the associations of MetS and metabolically unhealthy obesity (MUHO) with cancer mortality in a Japanese population.

### Methods

We used data from the Japan Multi-Institutional Collaborative Cohort Study. The study population consisted of 28,554 eligible subjects (14,103 men and 14,451 women) aged 35–69 years. MetS was diagnosed based on the criteria of the National Cholesterol Education Program Adult Treatment Panel III (NCEP-ATP III) and the Japan Society for the Study of Obesity (JASSO), using the body mass index instead of waist circumference. The Cox proportional hazards analysis was used to estimate adjusted hazard ratios (HR) and 95% confidence intervals (CI) for total cancer mortality in relation to MetS and its components.

**Data Availability Statement:** The data used in the present study cannot be made publicly available, because the participants in the J-MICC Study have not given informed consent for public data sharing, and the ethics committee of the Tokushima University Hospital does not approve data sharing. At this time, interested researchers cannot receive access to the data. However, interested researchers can contact the Administration Office of the ethics committee of the Nagoya University Graduate School of Medicine (iga-shinsa@adm. nagoya-u.ac.jp). After receiving a request, informed consent from the participants will be sought.

**Funding:** This work was supported by a Grants-in-Aid for Scientific Research on Priority Areas of Cancer (No. 17015018), Innovative Areas (No. 221S0001) and JSPS KAKENHI Grant (No. 16H06277) from the Japanese Ministry of Education, Culture, Sports, Science and Technology, awarded to KW, and by a JSPS KAKENHI Grant (18K10086) from the Japanese Ministry of Education, Culture, Sports, Science and Technology, awarded to KA.

**Competing interests:** The authors have declared that no competing interests existed.

Additionally, the associations of obesity and the metabolic health status with cancer mortality were examined.

## Results

During an average 6.9-year follow-up, there were 192 deaths from cancer. The presence of MetS was significantly correlated with increased total cancer mortality when the JASSO criteria were used (HR = 1.51, 95% CI 1.04–2.21), but not when the NCEP-ATP III criteria were used (HR = 1.09, 95% CI 0.78–1.53). Metabolic risk factors, elevated fasting blood glucose, and MUHO were positively associated with cancer mortality ($P < 0.05$).

## Conclusion

MetS diagnosed using the JASSO criteria and MUHO were associated with an increased risk of total cancer mortality in the Japanese population.

## Introduction

Metabolic syndrome (MetS) is characterized by the clustering of several cardiovascular risk factors, such as abdominal obesity, high blood pressure, dyslipidemia, and high blood glucose levels [1, 2]. MetS is associated with increased risks of the future development of type 2 diabetes and cardiovascular diseases [3, 4], and is currently a major public health problem throughout the world. Cancer mortality is a key measure of cancer's impact on health and is rapidly growing in both developed and developing countries [5]. The International Agency for Research on Cancer based on 20 world regions estimated that there were 9.6 million cancer deaths in 2018.

Epidemiologic data on the association between MetS and cancer mortality are inconsistent [6–8]. For example, a cohort study in Japan reported that MetS was associated with an increased risk of cancer mortality in women [6]. In Korea, MetS was reported to be a risk factor for cancer-related death among men [7]. On the other hand, a recent study by Iseki et al., which followed 664,926 Japanese adults for approximately 7 years, did not find any correlation between MetS and death from cancer [8].

Recently, the concepts of "metabolically healthy obesity" (MHO) and "metabolically unhealthy obesity" (MUHO) were proposed [9, 10]. The classification of obesity into MUHO and MHO phenotypes, which is based on the presence/absence of cardio-metabolic risk factors, may be useful to identify a subgroup of obese subjects at high/low risk of developing chronic diseases. Previous cohort studies reported that MUHO was associated with higher risks of cardiovascular diseases and all-cause mortality than MHO [9, 10].

The aim of the present study was to examine whether MetS and its components were associated with total cancer mortality in Japan Multi-Institutional Collaborative Cohort (J-MICC) Study. We also examined the risk of cancer mortality according to obesity and the metabolic health status.

## Materials and methods

### Study design and subjects

The current study was a serial prospective population-based cohort analysis using data from J-MICC Study. Details of J-MICC Study were described in a recent report [11]. In April 2005,

J-MICC Study was initiated under a population-based cohort study design, and it is managed by 14 research sites to examine gene–environment interactions in lifestyle-related diseases, including cancers, among Japanese.

The research protocol was approved by the ethics committee of Nagoya University Graduate School of Medicine (IRB No. 2010-0939-7), Aichi Cancer Center Research Institute (IRB No. 2016–2–10), Tokushima University Hospital (IRB No. 466–8), and all other institutions participating in J-MICC Study. During the survey, participants were informed that their participation was voluntary and written informed consent was obtained from all participants.

At seven study sites that used the same questionnaires and measured fasting blood glucose levels, 51,538 subjects participated (data set of Version 2020.12.18). We excluded study subjects who had a history of cancer, myocardial infarction, or stroke (n = 4,001), and lacked data on the follow-up period (n = 3), history of cancer, myocardial infarction, or stroke (n = 3,550), smoking habit (n = 22), alcohol drinking habit (n = 25), body mass index (BMI), systolic blood pressure (SBP), diastolic blood pressure (DBP), triglycerides (TG), high-density lipoprotein cholesterol (HDL-C), fasting blood glucose (n = 5,805), physical activity (n = 1,687), or the use of antihypertensive or hypoglycemic agents (n = 14). Finally, 28,554 participants (14,103 men and 14,451 women) were included in the present analysis (Fig 1).

## Questionnaire and covariates

Subjects were asked to fill out a self-administered questionnaire that included questions about age, sex, educational background, smoking habits, alcohol consumption, and exercise habits. Medical history was assessed by recording hypertension, diabetes, cardiovascular diseases, cancer, and other diseases. Educational background was categorized as ≤9 years, 10–15 years, ≥16 years, and unknown. Smoking habit was categorized as current, past, and never smokers. Alcohol drinking was categorized as current, past, and never drinkers. Exercise during leisure time was estimated by multiplying the frequency (5 categories from never to ≥5 times/week) and average duration (6 categories from ≤30 minutes to ≥4 hours) of low (such as walking, hiking, and golf at 3.4 metabolic equivalents (METs)), moderate (such as jogging, swimming, skiing, and dancing at 7.0 METs), and powerful intensity exercises (such as marathon running, intense ball games, and combat sports at 10.0 METs). The three levels of exercise were summarized and presented as MET-hours/week. Height and weight were measured, and BMI was calculated by weight in kg divided by height in meters squared ($kg/m^2$).

## Diagnosis of metabolic syndrome and metabolic health status

MetS was diagnosed according to the criteria of the National Cholesterol Education Program Adult Treatment Panel III (NCEP-ATP III) [12], using BMI instead of waist circumference (WC). Participants were diagnosed as having MetS when at least three of the following five conditions were satisfied: (i) BMI ≥25 $kg/m^2$; (ii) SBP ≥130 mmHg and/or DBP ≥85 mmHg or the use of antihypertensive medication; (iii) serum TG level ≥150 mg/dL; (iv) serum HDL-C level <40 mg/dL for men and <50 mg/dL for women; and (v) fasting blood glucose level ≥100 mg/dL or use of hypoglycemic agents. We also applied the criteria of the Japan Society for the Study of Obesity (JASSO) [13], using BMI instead of WC. By these criteria, study subjects were diagnosed as having MetS when obesity (≥ 25 $kg/m^2$) and two or more of the following conditions were satisfied: SBP ≥130 mmHg and/or DBP ≥85 mmHg or the use of antihypertensive medication; elevated TG (≥150 mg/dL) or reduced HDL-C (<40 mg/dL); and elevated fasting glucose (≥110 mg/dL) or the use of antidiabetic medication.

Normal weight participants (BMI <25 $kg/m^2$) were classified as having a metabolically unhealthy normal weight or metabolically healthy normal weight (MHNW) (≥1 or no

Baseline data in 14 different regions to examine gene–environment interactions in lifestyle-related diseases, including cancers, among Japanese with a mean follow-up of 9 years.

7 study sites which used the same questionnaires and measured fasting blood glucose levels. 51,538 subjects participated.

Excluded:

- Previous history of cancer, myocardial infarction, or stroke (n=4,001).
- Lacking data on the follow-up period, smoking, alcohol drinking, daily physical activity, body mass index, systolic and diastolic blood pressure, high-density lipoprotein cholesterol, triglycerides, blood glucose, and the use of blood pressure and blood glucose medication (n=18,983).

28,554 participants (14,103 men and 14,451 women) were eligible for the analysis.

**Fig 1. A flowchart showing the process for selecting the study subjects.**

components of MetS, respectively). Similarly, obese subjects (BMI $\geq$25 kg/m$^2$) were classified as MUHO or MHO ($\geq$1 or no components of MetS other than BMI, respectively).

### Follow-up

The causes of death were confirmed by death certificates, after obtaining permission from the Japanese Ministry of Health, Labour and Welfare. Study subjects who had moved out of the study area were treated as censored cases. The follow-up period for each subject was calculated as the time from the date of health examination to the occurrence of death, transfer, or the end of follow-up (2016 or 2017), whichever came first. Cancer death was classified according to the International Classification of Diseases, 10th revision. Mortality from cancer was defined by codes C021-97. During a mean follow-up of 6.9 years, death was recorded for 396 subjects, of which 192 were from cancer.

### Statistical analysis

Background characteristics of participants were compared according to the presence or absence of MetS. Continuous variables are expressed as the median (25%, 75%), and categorical variables are expressed as numbers and proportions (%). The Wilcoxon's rank sum test and Chi-square test were used to examine the differences in the characteristics of study subjects according to MetS.

The Cox proportional hazards regression model was applied to estimate multivariate adjusted hazard ratios (HR) and 95% confidence intervals (CI) for the association of MetS, number of metabolic risk factors, and each of its individual components with total cancer mortality. We also analyzed the associations of metabolic health phenotypes with cancer-related mortality. Model 1 was adjusted for age (continuous; years), menopausal status (men, premenopausal women, postmenopausal women, and missing), research sites (7 sites), and educational background (categorical; $\leq$9 years, 10–15 years, $\geq$16 years, and unknown); Model 2 was additionally adjusted for smoking status (current, past, and never), drinking status (current, past, and never), and physical activity level (quartiles). The proportional hazards assumption was checked using 3 methods: (1) drawing the log-negative-log plot of survival function; (2) testing the significance of the product term of exposure variable and log(time); and (3) plotting Schoenfeld residuals against time. All statistical analyses were performed using the statistical software package SAS version 9.4 (SAS Institute, Cary, NC, USA). Statistical tests were based on two-sided probabilities, and *P*-value of less than 0.05 was considered significant.

### Results

Table 1 shows descriptive data on the baseline characteristics of participants with and without MetS. Among 28,554 participants, 17.0% of the total subjects were diagnosed as having MetS. Those with MetS were more likely to be men and older. There were no significant differences in leisure-time physical activity levels between participants with and without MetS. Relative to those without MetS, participants with MetS showed higher rates of education $\leq$9 years, current smokers, current drinkers, obesity, and postmenopausal women. The proportions of those who had self-reported histories of colorectal polyps, fatty liver, high blood pressure, diabetes and dyslipidemia, and medication for high blood pressure, diabetes, and high blood cholesterol were higher, while histories of chronic gastritis and medication for constipation were less prevalent among participants with MetS than in those without MetS.

Results of total cancer mortality associated with MetS as well as the number of its components are displayed in Table 2. When the modified NCEP-ATP III criteria were used, there was no significant correlation between MetS and total cancer mortality [multivariate-adjusted

**Table 1. Background characteristics of participants according to metabolic syndrome status.**

| Characteristics[b] | Metabolic syndrome[a] | | P-value[c] |
|---|---|---|---|
| | **No** | **Yes** | |
| | **(n = 23850)** | **(n = 4704)** | |
| Age (years) | 55 (46, 62) | 58 (50, 64) | <0.0001 |
| Body mass index (kg/m²) | 22.2 (20.5, 24.0) | 26.3 (25.1, 28.2) | <0.0001 |
| Systolic blood pressure (mmHg) | 122 (110, 133) | 136 (129, 146) | <0.0001 |
| Diastolic blood pressure (mmHg) | 75 (68, 82) | 84 (78, 90) | <0.0001 |
| Triglycerides (mg/dL) | 85 (63, 117) | 171 (122, 228) | <0.0001 |
| HDL-cholesterol (mg/dL) | 65 (55, 77) | 49.1 (42.8, 59.6) | <0.0001 |
| Fasting plasma glucose (mg/dL) | 92 (87, 98) | 105 (98, 116) | <0.0001 |
| Exercise during leisure time (MET-hours/week) | 5.6 (0.43, 17.93) | 6.45 (0.43, 17.85) | 0.79 |
| **Sex** | | | |
| Male | 10928 (45.8) | 3175 (67.5) | <0.0001 |
| Female | 12922 (54.2) | 1529 (32.5) | |
| **Educational background (years)** | | | |
| ≤9 | 2637 (11.1) | 806 (17.1) | <0.0001 |
| 10–15 | 15317 (64.2) | 2732 (58.1) | |
| ≥16 | 5767 (24.2) | 1134 (24.1) | |
| Unknown | 129 (0.5) | 32 (0.7) | |
| **Smoking habit** | | | |
| Current | 3785 (15.9) | 951 (20.2) | <0.0001 |
| Past | 5296 (22.2) | 1501 (31.9) | |
| Never | 14769 (61.9) | 2252 (47.9) | |
| **Alcohol drinking** | | | |
| Current | 13523 (56.7) | 2975 (63.2) | <0.0001 |
| Past | 389 (1.6) | 70 (1.5) | |
| Never | 9938 (41.7) | 1659 (35.3) | |
| **Obesity status** | | | |
| Non-obese | 20429 (85.7) | 1109 (23.6) | <0.0001 |
| Obese | 3421 (14.3) | 3595 (76.4) | |
| **Menopausal status of women** | | | |
| Premenopausal | 5299 (41.0) | 308 (20.1) | <0.0001 |
| Postmenopausal | 7553 (58.5) | 1214 (79.4) | |
| Missing | 70 (0.5) | 7 (0.5) | |
| **Medical history** | | | |
| Gastric ulcer | 2843 (11.9) | 557 (11.9) | 0.65 |
| Colorectal polyps | 1988 (8.4) | 538 (11.4) | <0.0001 |
| Chronic gastritis | 2805 (11.8) | 454 (9.7) | <0.0001 |
| Hepatitis B | 285 (1.2) | 65 (1.4) | 0.29 |
| Hepatitis C | 188 (0.8) | 40 (0.9) | 0.66 |
| Fatty liver | 1650 (6.9) | 956 (20.5) | <0.0001 |
| Asthma | 1477 (6.2) | 315 (6.7) | 0.40 |
| High blood pressure | 3417 (14.4) | 1928 (41.1) | <0.0001 |
| Diabetes | 854 (3.6) | 675 (14.4) | <0.0001 |
| Dyslipidemia | 3074 (12.9) | 1227 (26.3) | <0.0001 |
| **Medication** | | | |
| High blood pressure | 2795 (11.7) | 1707 (36.3) | <0.0001 |
| Diabetes | 506 (2.1) | 500 (10.6) | <0.0001 |

*(Continued)*

**Table 1.** (Continued)

| Characteristics[b] | Metabolic syndrome[a] | | P-value[c] |
|---|---|---|---|
| | No | Yes | |
| | (*n* = 23850) | (*n* = 4704) | |
| High blood cholesterol | 1867 (7.8) | 764 (16.2) | <0.0001 |
| Sleeping pills | 791(3.3) | 181(3.9) | 0.07 |
| Antipyretic | 679 (2.9) | 130 (2.8) | 0.75 |
| Laxative | 916 (3.8) | 117 (2.5) | <0.0001 |

HDL, high-density lipoprotein; MET, metabolic equivalent.

[a] Diagnosed using the National Cholesterol Education Program Adult Treatment Panel III criteria with modification.

[b] Median (25%, 75%) or number of subjects (%).

[c] Wilcoxon's rank sum test or Chi-square test.

HR (95% CI): 1.09 (0.78, 1.53)]. The trend regarding the association between the number of abnormal components of MetS and cancer mortality was marginally significant (*P*-trend = 0.06). There was a marginally significant correlation between obesity and total cancer mortality. In addition, high fasting blood glucose was associated with increased cancer-related death, with significance [HR (95% CI): 1.41, (1.05, 1.89)]. In Table 3, participants with MUHO had a significantly higher risk of dying from cancer compared with those with MHNW [HR (95% CI): 1.76, (1.10, 2.80)]. However, when the presence of ≥ two components of MetS (other than BMI) was used to define a metabolically unhealthy status, cancer mortality among the MUHO group was not significantly increased [HR (95% CI): 1.42, 0.95, 2.10)] (S1 Table).

When the modified criteria of JASSO were used, MetS was significantly correlated with increased mortality from cancer [HR (95% CI): 1.51, (1.04, 2.21)] (Table 4). The results for the number of metabolic abnormalities and each component of MetS, MHO, and MUHO, were essentially similar to those in Tables 2–5.

Finally, the associations between MetS and mortality from site-specific cancers were examined (esophagus, stomach, colorectum, liver, gallbladder and biliary tract, pancreas, and lung). Point estimates of HR were higher than 1.0 for stomach, colorectum, liver, and pancreas. However, significantly increased HR was observed only for colorectal cancer [HR (95% CI): 2.95, (1.04, 8.40), JASSO criteria].

## Discussion

### MetS and cancer mortality

In the present study, MetS was significantly correlated with an increased risk of total cancer mortality when the JASSO criteria were used, but not when the NCEP-ATP III criteria were used. To our knowledge, at least six previous cohort studies examined the association between MetS and total cancer mortality [6–8, 14–16], but the results were inconsistent. Two U.S. studies (one study [14] included only men) observed significant positive correlations [14, 15], a Korean study observed a significant positive correlation only in men [7], and a Japanese study found a positive association only in women [6]. On the other hand, two studies recently performed in the U.S. and Japan found no association [8, 16]. The reason for this inconsistency could not be explained by the differences in country (U.S., where the prevalence of obesity is higher than in Asian countries, Korea and Japan), the criteria used for the diagnosis of MetS (NCEP-ATP III and JASSO criteria), the number of subjects (7,028–664,926), or the potential confounders adjusted for. In our study, a significant correlation was observed only when the

**Table 2. Hazard ratios and 95% confidence intervals for total cancer mortality in relation to metabolic syndrome and its components.**

| | Presence | Participants | Cancer deaths | Person-years | Crude mortality (person/1000 person-years) | HR[a] (95% CI) | HR[b] (95% CI) |
|---|---|---|---|---|---|---|---|
| Metabolic syndrome[c] | No | 23850 | 145 | 164479 | 0.88 | 1 | 1 |
| | Yes | 4704 | 47 | 33958 | 1.38 | 1.09 (0.78, 1.53) | 1.09 (0.78, 1.53) |
| Number of metabolic risk factors | 0 | 9042 | 27 | 62072 | 0.43 | 1 | 1 |
| | 1 | 8516 | 57 | 58541 | 0.97 | 1.45 (0.91, 2.30) | 1.44 (0.90, 2.29) |
| | 2 | 6292 | 61 | 43866 | 1.39 | 1.73 (1.09, 2.75) | 1.74 (1.10, 2.77) |
| | ≥3 | 4704 | 47 | 33958 | 1.38 | 1.58 (0.97, 2.57) | 1.58 (0.97, 2.58) |
| | | | | | | P-trend = 0.06 | P-trend = 0.06 |
| Obesity | No | 21538 | 126 | 147218 | 0.86 | 1 | 1 |
| | Yes | 7016 | 66 | 51219 | 1.29 | 1.26 (0.93, 1.71) | 1.30 (0.96, 1.77) |
| High blood pressure | No | 15335 | 73 | 108016 | 0.68 | 1 | 1 |
| | Yes | 13219 | 119 | 90421 | 1.32 | 1.16 (0.85, 1.56) | 1.18 (0.87, 1.60) |
| Elevated triglycerides | No | 22905 | 153 | 159042 | 0.96 | 1 | 1 |
| | Yes | 5649 | 39 | 39395 | 0.99 | 0.83 (0.58, 1.18) | 0.80 (0.56, 1.14) |
| Low HDL-cholesterol | No | 26109 | 176 | 180892 | 0.97 | 1 | 1 |
| | Yes | 2445 | 16 | 17545 | 0.91 | 0.96 (0.57, 1.60) | 0.94 (0.56, 1.58) |
| Elevated blood glucose | No | 20065 | 100 | 137146 | 0.73 | 1 | 1 |
| | Yes | 8489 | 92 | 61291 | 1.50 | 1.42 (1.06, 1.90) | 1.41 (1.05, 1.89) |

HR, hazard ratio; CI, confidence interval; HDL, high-density lipoprotein.

[a] Adjusted for age, menopausal status (men, premenopausal women, postmenopausal women, and missing), research sites, and educational background.

[b] Additionally adjusted for smoking habit (three categories), drinking habit (three categories), and physical activity level (quartiles).

[c] Diagnosed using the National Cholesterol Education Program Adult Treatment Panel III criteria with modification.

**Table 3. Hazard ratios and 95% confidence intervals for total cancer mortality in relation to metabolically healthy status and body mass index.**

| Group | Participants | Cancer deaths | Person-years | Crude mortality (persons/1000 person-years) | HR[a] (95% CI) | HR[b] (95% CI) |
|---|---|---|---|---|---|---|
| Metabolically healthy normal weight | 9042 | 27 | 62072 | 0.43 | 1 | 1 |
| Metabolically healthy obese | 1017 | 6 | 7554 | 0.79 | 1.67 (0.69, 4.05) | 1.71 (0.70, 4.15) |
| Metabolically unhealthy normal weight | 12496 | 99 | 85147 | 1.16 | 1.50 (0.97, 2.32) | 1.48 (0.96, 2.28) |
| Metabolically unhealthy obese | 5999 | 60 | 43665 | 1.37 | 1.72 (1.08, 2.74) | 1.76 (1.10, 2.80) |

HR, hazard ratio; CI, confidence interval.

[a] Adjusted for age, menopausal status (men, premenopausal women, postmenopausal women, and missing), research sites, and educational background.

[b] Additionally adjusted for smoking habit (three categories), drinking habit (three categories), and physical activity level (quartiles).

**Table 4. Hazard ratios and 95% confidence intervals for total cancer mortality in relation to metabolic syndrome and its components diagnosed using the criteria of Japan Society for the Study of Obesity with modification.**

| | Presence | Participants | Cancer deaths | Person-years | Crude mortality (person/1000 person-years) | HR[a] (95% CI) | HR[b] (95% CI) |
|---|---|---|---|---|---|---|---|
| Metabolic syndrome | No | 26000 | 158 | 180176 | 0.88 | 1 | 1 |
| | Yes | 2554 | 34 | 18261 | 1.86 | 1.50 (1.03, 2.19) | 1.51 (1.04, 2.21) |
| Number of metabolic risk factors[c] | 0 | 10520 | 34 | 72793 | 0.47 | 1 | 1 |
| | 1 | 9299 | 67 | 64001 | 1.05 | 1.45 (0.95, 2.20) | 1.46 (0.96, 2.22) |
| | 2 | 5507 | 53 | 38642 | 1.37 | 1.63 (1.05, 2.53) | 1.65 (1.06, 2.56) |
| | ≥3 | 3228 | 38 | 23001 | 1.65 | 1.78 (1.11, 2.88) | 1.79 (1.11, 2.89) |
| | | | | | | P-trend = 0.01 | P-trend = 0.01 |
| Obesity | No | 21538 | 126 | 147218 | 0.86 | 1 | 1 |
| | Yes | 7016 | 66 | 51219 | 1.29 | 1.26 (0.93, 1.71) | 1.30 (0.96, 1.77) |
| High blood pressure | No | 15335 | 73 | 108016 | 0.68 | 1 | 1 |
| | Yes | 13219 | 119 | 90421 | 1.32 | 1.15 (0.85, 1.56) | 1.18 (0.87, 1.60) |
| Elevated triglycerides | No | 22905 | 153 | 159042 | 0.96 | 1 | 1 |
| | Yes | 5649 | 39 | 39395 | 0.99 | 0.83 (0.58, 1.18) | 0.80 (0.56, 1.14) |
| Low HDL-cholesterol | No | 27330 | 179 | 189628 | 0.94 | 1 | 1 |
| | Yes | 1224 | 13 | 8809 | 1.48 | 1.24 (0.70, 2.19) | 1.21 (0.69, 2.15) |
| Elevated triglycerides or low HDL-cholesterol | No | 22464 | 145 | 155897 | 0.93 | 1 | 1 |
| | Yes | 6090 | 47 | 42540 | 1.10 | 0.95 (0.68, 1.32) | 0.91 (0.65, 1.28) |
| Elevated blood glucose | No | 24807 | 136 | 171850 | 0.79 | 1 | 1 |
| | Yes | 3747 | 56 | 26587 | 2.11 | 1.77 (1.29, 2.44) | 1.74 (1.27, 2.39) |

HR, hazard ratio; CI, confidence interval; HDL, high-density lipoprotein.

[a] Adjusted for age, menopausal status (men, premenopausal women, postmenopausal women, and missing), research sites, and educational background.

[b] Additionally adjusted for smoking habit (three categories), drinking habit (three categories), and physical activity level (quartiles).

[c] Obesity +high blood pressure +elevated triglycerides +low HDL-cholesterol +elevated blood glucose.

JASSO criteria were used. This may be because obesity was a prerequisite component of MetS in the JASSO criteria, and HR associated with high blood glucose was higher when 110 mg/dL was used as a cut-off level.

## Number of MetS components and cancer mortality

In our study, cancer mortality was higher in participants with 2 and ≥3 components of MetS, respectively, compared with those with no component (modified JASSO criteria, Table 4). In a study conducted in the U.S., Gathirua-Mwangi et al., reported that those who had 3, 4, and 5 abnormal components of MetS had a 28, 24, and 87%, respectively, higher risk of dying from cancer than those with 0–2 abnormal components [15]. The Jichi Medical School (JMS) Cohort Study, which followed 4,495 men and 7,028 women for 18.5 years, reported that an increase in the number of MetS components was associated with increased cancer mortality

**Table 5. Hazard ratios and 95% confidence intervals for total cancer mortality in relation to metabolically healthy status and body mass index (criteria of Japan Society for the Study of Obesity were used, with modification).**

| Group | Participants | Site-specific cancer deaths | Person-years | Crude mortality (person/1000 person-years) | HR[a] (95% CI) | HR[b] (95% CI) |
|---|---|---|---|---|---|---|
| Metabolically healthy normal weight | 10520 | 34 | 72793 | 0.47 | 1 | 1 |
| Metabolically healthy obese | 1428 | 10 | 10815 | 0.92 | 1.82 (0.90, 3.70) | 1.88 (0.93, 3.82) |
| Metabolically unhealthy normal weight | 11018 | 92 | 74425 | 1.24 | 1.50 (1.00, 2.24) | 1.49 (0.99, 2.23) |
| Metabolically unhealthy obese | 5588 | 56 | 40404 | 1.39 | 1.65 (1.06, 2.56) | 1.69 (1.09, 2.63) |

HR, hazard ratio; CI, confidence interval.

[a] Adjusted for age, menopausal status (men, premenopausal women, postmenopausal women, and missing), research sites, and educational background.

[b] Additionally adjusted for smoking habit (three categories), drinking habit (three categories), and physical activity level (quartiles).

among Japanese [6]. Results of earlier studies performed in the U.S. [14, 16] and Korea were essentially similar, although *P* for trend was not given in the Korean study [7]. The current results corroborated and expanded further the dose-response relationship between the number of MetS components and cancer mortality.

## Obesity, blood glucose, and cancer mortality

Our results showed that elevated blood glucose was associated with a 1.41-times increased risk of cancer death, being in accordance with the results of previous studies. All five earlier cohort studies on each component of MetS reported increased HRs associated with high fasting blood glucose [6, 7, 14–16]. A Korean study on MUHO also reported that diabetes, and diabetes combined with hypertension, but not dyslipidemia, were associated with a significantly increased risk of cancer mortality [17]. Furthermore, in a meta-analysis of 19 Asian prospective studies involving 771,000 study subjects, self-reported diabetes was associated with a 26% increased risk of death from any cancer in Asians [18]. The biological mechanisms linking obesity/high blood glucose to cancer have been reviewed in detail [19, 20]. Insulin resistance, characterized by high insulin secretion by beta cells to compensate for high blood glucose, is an underlying key condition of MetS. Hyperinsulinemia increases insulin-like growth factor (IGF)-1 production in the liver, and decreases the levels of IGF binding proteins, leading to increased bioavailable IGF-1 and IGF-2 levels. In the presence of hyperinsulinemia and high IGF levels, cancer cells upregulate insulin and IGF-1 receptors, resulting in the stimulation of signaling pathways that are closely related to mitogenesis, cell growth, and migration [19, 20]. In a case-cohort study performed in Japan, plasma C-peptide concentrations were associated with significantly increased risks of all-cancers and cancers of five sites [21]. Obesity-associated oxidative stress and inflammation are also involved in the development/progression of various cancers [22].

## MHO, MUHO, and cancer mortality

The current study revealed that cancer mortality among the MUHO group was 1.76 times higher than that in the MHNW group. JMS Cohort Study reported a 3.3-fold increased risk of cancer mortality in MUHO subjects (BMI≥30 kg/m$^2$) when compared with MHNW subjects (25> BMI ≥18.5 kg/m$^2$) [23]. In the U.K. Biobank cohort, the MUHO group (BMI≥30 kg/m$^2$) had a significantly increased incidence rate of 10 cancers compared with the MHNW

group (25> BMI $\geq$18.5 kg/m$^2$) [24]. Other studies reported increased risks of obesity-related cancers, such as colorectal [25], pancreatic [26], and postmenopausal breast cancer [27], among MHUO than MHNW subjects. In contrast, another study using a nationwide dataset of the Korean population reported that individuals with the MUHO (BMI $\geq$25 kg/m$^2$) phenotype did not show increased cancer mortality compared with the MHNW phenotype (BMI <25 kg/m$^2$) [17]. It should be pointed out that the cut-off levels for BMI to define obesity/normal weight differed among the studies, and there was no standard definition of metabolic abnormalities (NCEP-ATP III criteria [26, 27] or high BMI plus $\geq$ one [17] or $\geq$ two [23–25] abnormal components of high blood pressure, high serum TG levels, low serum HDL-C levels, and high blood glucose levels). In our study, when $\geq$ two components of MetS were used to define a metabolically unhealthy status, cancer mortality among the MUHO group was not significantly increased. Regarding MHO, the U.K. Biobank cohort observed a significantly increased risk of five cancers among the MHO group (BMI $\geq$30 kg/m$^2$) than the MHNW group [24]. In our study, the point estimate of HR among the MHO subjects was 1.71, and in the JMS cohort study, the corresponding figure was 1.8, but these results were not significant, probably because of the small number of cancer deaths. Thus, the associations of MUHO and MHO with cancer mortality need further investigation.

## Strengths and limitations

Major strengths of the present study include the following: we examined the risk of all-cancer mortality associated with MetS and its components in a general Japanese population recruited from various regions. In addition, various potential confounders were adjusted using multivariate modelling. On the other hand, several limitations of the present study are worth mentioning. First, the diagnosis of MetS was based on a single measurement only at the baseline, and lacked updates of the MetS status and its components during follow-up. Therefore, it was difficult to assess the impact of changes in the MetS status over time on cancer mortality. Second, the current study used BMI instead of WC for the diagnosis of MetS because data on WC were not available for every study site. WC is an indicator of visceral fat mass, while BMI is an indicator of general body fat mass. In previous large-scale prospective studies, both high WC and high BMI were associated with increased risks of cancers [24], but high WC was associated with several cancers independent of a high BMI [28]. Thus, although BMI closely correlates with WC, the use of BMI instead of WC may have had some influences on our results. Third, as information on smoking and drinking habits, leisure-time exercise, and other background characteristics was obtained using a self-reported questionnaire, misclassifications/measurement errors may be inevitable. Fourth, there were relatively small numbers of total (192 subjects) and site-specific (S2 Table) cancer deaths. Therefore, in this study population, it was difficult to examine the associations of MetS and its components with site-specific cancer mortality.

## Conclusions

In conclusion, a significant correlation between MetS and the overall cancer death rate was observed when the modified JASSO criteria were used, and an increasing number of MetS components was associated with an increased risk of cancer mortality. High blood glucose was associated with an increased risk of all-cancer death. Moreover, our results suggest that cancer mortality was higher among MUHO than MHNW subjects. The findings of the present study provide additional evidence useful for the prevention and management of cancer in participants with MetS and MUHO, especially those with elevated blood glucose. Further studies are needed to confirm the influence of MetS, MUHO, and MHO on the risk of cancer mortality.

## Supporting information

**S1 Table. Hazard ratios and 95% confidence intervals for total cancer mortality in relation to metabolically healthy status and body mass index (when presence of $\geq$ two components of metabolic syndrome other than body mass index was used to define a metabolically unhealthy status).** HR, hazard ratio; CI, confidence interval. [a] Adjusted for age, menopausal status (men, premenopausal women, postmenopausal women, and missing), research sites, and educational background. [b] Additionally adjusted for smoking habit (three categories), drinking habit (three categories), and physical activity level (quartiles).
(XLSX)

**S2 Table. The number of site-specific cancer deaths according to metabolic syndrome status.**
(XLSX)

## Acknowledgments

We are deeply indebted to all participants in the baseline survey of J-MICC Study. We thank the following researchers for providing us with a food frequency questionnaire and program to estimate nutrient intake: Shinkan Tokudome (formerly National Institute of Health and Nutrition and Nagoya City University), Yuko Tokudome (Nagoya University of Arts and Sciences), Masato Ikeda (University of Occupational and Environmental Health), and Shinzo Maki (Aichi Prefectural Dietetic Association). We also thank the former principal investigators, Nobuyuki Hamajima and Hideo Tanaka, for their great efforts and contributions to J-MICC Study.

## Author Contributions

**Conceptualization:** Tien Van Nguyen, Kokichi Arisawa.

**Data curation:** Tien Van Nguyen, Kokichi Arisawa, Sakurako Katsuura-Kamano, Masashi Ishizu, Mako Nagayoshi, Rieko Okada, Asahi Hishida, Takashi Tamura, Megumi Hara, Keitaro Tanaka, Daisaku Nishimoto, Keiichi Shibuya, Teruhide Koyama, Isao Watanabe, Sadao Suzuki, Takeshi Nishiyama, Kiyonori Kuriki, Yasuyuki Nakamura, Yoshino Saito, Hiroaki Ikezaki, Jun Otonari, Yuriko N. Koyanagi, Keitaro Matsuo, Haruo Mikami, Miho Kusakabe, Kenji Takeuchi, Kenji Wakai.

**Formal analysis:** Tien Van Nguyen, Kokichi Arisawa.

**Funding acquisition:** Kokichi Arisawa.

**Investigation:** Sakurako Katsuura-Kamano, Masashi Ishizu.

**Methodology:** Tien Van Nguyen, Kokichi Arisawa.

**Project administration:** Kenji Takeuchi, Kenji Wakai.

**Resources:** Keitaro Tanaka, Teruhide Koyama, Sadao Suzuki, Kiyonori Kuriki, Hiroaki Ikezaki, Keitaro Matsuo, Haruo Mikami, Kenji Takeuchi, Kenji Wakai.

**Supervision:** Keitaro Tanaka, Teruhide Koyama, Sadao Suzuki, Kiyonori Kuriki, Hiroaki Ikezaki, Keitaro Matsuo, Haruo Mikami, Kenji Takeuchi, Kenji Wakai.

**Writing – original draft:** Tien Van Nguyen, Kokichi Arisawa.

**Writing – review & editing:** Tien Van Nguyen, Kokichi Arisawa.

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
