## [Decision Letter · Decision Letter 0]

24 Jan 2022

PONE-D-21-35995Associations of metabolic syndrome and metabolically unhealthy obesity with cancer mortality: Results of prospective cohort study in Japanese populationPLOS ONE

Dear Dr. Arisawa,

Thank you for submitting your manuscript to PLOS ONE. After careful consideration, we feel that it has merit but does not fully meet PLOS ONE’s publication criteria as it currently stands. Therefore, we invite you to submit a revised version of the manuscript that addresses the points raised during the review process.

We look forward to receiving your revised manuscript.

Kind regards,

Venkata Naga Srikanth Garikipati, PhD

Academic Editor

PLOS ONE

Journal Requirements:

6. One of the noted authors is a group or consortium “Japan Multi-institutional Collaborative Cohort (J-MICC) Study Group.” In addition to naming the author group, please list the individual authors and affiliations within this group in the acknowledgments section of your manuscript. Please also indicate clearly a lead author for this group along with a contact email address.

7. We note that you have included the phrase “data not shown” in your manuscript. Unfortunately, this does not meet our data sharing requirements. PLOS does not permit references to inaccessible data. We require that authors provide all relevant data within the paper, Supporting Information files, or in an acceptable, public repository. Please add a citation to support this phrase or upload the data that corresponds with these findings to a stable repository (such as Figshare or Dryad) and provide and URLs, DOIs, or accession numbers that may be used to access these data. Or, if the data are not a core part of the research being presented in your study, we ask that you remove the phrase that refers to these data.

Reviewers' comments:

Reviewer's Responses to Questions

**Comments to the Author**

1. Is the manuscript technically sound, and do the data support the conclusions?

Reviewer #1: Yes

Reviewer #2: Yes

2. Has the statistical analysis been performed appropriately and rigorously? 

Reviewer #1: No

Reviewer #2: I Don't Know

3. Have the authors made all data underlying the findings in their manuscript fully available?

Reviewer #1: Yes

Reviewer #2: Yes

4. Is the manuscript presented in an intelligible fashion and written in standard English?

Reviewer #1: No

Reviewer #2: Yes

5. Review Comments to the Author

Reviewer #1: The authors examined the associations of MetS and metabolically unhealthy obesity (MUHO) with cancer

mortality in a Japanese population.. Although the finding are potential, reviewer has few concerns:

- Table 1 is broken. Reviewer couldn't read the table.

- Consult the statistician to confirm the statistical tests used in the study.

- Check for the grammar and typo error.

- Please report the other medical underlying conditions and how they affect the finding? for instance asthma, diabetes.

- What were the other medications, patients were taking? Correlate them with the findings.

Reviewer #2: The author aimed to make a correlation with metabolic syndrome and cancer mortality, which in my view was successful. more and more studies of these sort need to be published reviewing the literature which gives a complete overview of the field for future scientists.

6. PLOS authors have the option to publish the peer review history of their article (what does this mean?). If published, this will include your full peer review and any attached files.

Reviewer #1: No

Reviewer #2: **Yes: **Ajay Palagani

---

## [Author Response · Author response to Decision Letter 0]

15 Mar 2022

To Editorial Office, PLOS ONE

Thank you for your e-mail dated January 24, 2022 and very useful comments raised by the reviewers. We have modified the manuscript according to the comments. The changes and responses to the reviewer’s comments are as follows.

 We hope that our revised version suitable for publication in the PLOS ONE.

Sincerely yours,

Kokichi Arisawa, MD, MSc, PhD

Department of Preventive Medicine,

Tokushima University Graduate School of Biomedical Sciences,

3-18-15, Kuramoto-cho, Tokushima 770-8503, Japan

（Responses to Editor and Reviewers）

Journal Requirements:

The manuscript was modified according to the PLOS One’s style requirements.

Tables were included as part of the text. Supplementary tables 1 and 2 were attached as supporting information files.

In the Methods section, full ethics statement was added, including the names of the IRBs, the numbers, and written informed consent.

The research protocol was approved by the ethics committee of Nagoya University Graduate School of Medicine (IRB No. 2010-0939-7), Aichi Cancer Center Research Institute (IRB No. 2016- 2-10), Tokushima University Hospital (IRB No. 466-8), and all other institutions participating in J-MICC Study. During the survey, participants were informed that their participation was voluntary and written informed consent was obtained from all participants.

Name and number of grants were added to the Funding Information section.

The data used in the present study cannot be made publicly available, because the participants in the J-MICC Study have not given informed consent for public data sharing, and the ethics committee of the Tokushima University Hospital does not approve data sharing. At this time, interested researchers cannot receive access to the data. However, interested researchers can contact the Administration Office of the ethics committee of the Nagoya University Graduate School of Medicine (iga-shinsa@adm.nagoya-u.ac.jp). After receiving a request, informed consent from the participants will be sought.

6. One of the noted authors is a group or consortium “Japan Multi-institutional Collaborative Cohort (J-MICC) Study Group.” In addition to naming the author group, please list the individual authors and affiliations within this group in the acknowledgments section of your manuscript. Please also indicate clearly a lead author for this group along with a contact email address.

The name of the Japan Multi-Institutional Collaborative Cohort (J-MICC) Study was deleted from the authors. Instead, it was included in the title.

7. We note that you have included the phrase “data not shown” in your manuscript. Unfortunately, this does not meet our data sharing requirements. PLOS does not permit references to inaccessible data. We require that authors provide all relevant data within the paper, Supporting Information files, or in an acceptable, public repository. Please add a citation to support this phrase or upload the data that corresponds with these findings to a stable repository (such as Figshare or Dryad) and provide and URLs, DOIs, or accession numbers that may be used to access these data. Or, if the data are not a core part of the research being presented in your study, we ask that you remove the phrase that refers to these data.

Supplementary table 1 was added and the phrase “data not shown” was deleted.

Captions for supporting information files were added to the end of the text.

Reference list was checked and modified.

Reviewers' comments:

Reviewer's Responses to Questions

Comments to the Author

1. Is the manuscript technically sound, and do the data support the conclusions?

Reviewer #1: Yes

Reviewer #2: Yes

2. Has the statistical analysis been performed appropriately and rigorously? 

Reviewer #1: No

Reviewer #2: I Don't Know

3. Have the authors made all data underlying the findings in their manuscript fully available?

Reviewer #1: Yes

Reviewer #2: Yes

4. Is the manuscript presented in an intelligible fashion and written in standard English?

Reviewer #1: No

Reviewer #2: Yes

5. Review Comments to the Author

Reviewer #1: The authors examined the associations of MetS and metabolically unhealthy obesity (MUHO) with cancer

mortality in a Japanese population.. Although the finding are potential, reviewer has few concerns:

- Table 1 is broken. Reviewer couldn't read the table.

Table 1 was included in the text.

- Consult the statistician to confirm the statistical tests used in the study.

Statistical methods used in this paper are commonly used ones, and our research team have sufficient experiences in epidemiologic research. In addition, data analysis was conducted by two independent persons, and it was confirmed that the figures in tables were identical.

- Check for the grammar and typo error.

The manuscript got English proofreading and errors were corrected.

- Please report the other medical underlying conditions and how they affect the finding? for instance asthma, diabetes.

- What were the other medications, patients were taking? Correlate them with the findings.

Other medications and histories of previous diseases were added to the text and the Table 1. These medications and previous diseases are as expected and may not have strong confounding effects on the results.

The proportions of those who had self-reported histories of colorectal polyps, fatty liver, high blood pressure, diabetes and dyslipidemia, and medication for high blood pressure, diabetes, and high blood cholesterol were higher, while histories of chronic gastritis and medication for constipation were less prevalent among participants with MetS than in those without MetS.

Reviewer #2: The author aimed to make a correlation with metabolic syndrome and cancer mortality, which in my view was successful. more and more studies of these sort need to be published reviewing the literature which gives a complete overview of the field for future scientists.

Thank you for your positive comments.

6. PLOS authors have the option to publish the peer review history of their article (what does this mean?). If published, this will include your full peer review and any attached files.

Do you want your identity to be public for this peer review? For information about this choice, including consent withdrawal, please see our Privacy Policy.

Reviewer #1: No

Reviewer #2: Yes: Ajay Palagani

---

## [Editor Report · Decision Letter 1]

24 May 2022

Associations of metabolic syndrome and metabolically unhealthy obesity with cancer mortality: The Japan Multi-Institutional Collaborative Cohort (J-MICC) Study

PONE-D-21-35995R1

Dear Dr.Arisawa,

We’re pleased to inform you that your manuscript has been judged scientifically suitable for publication and will be formally accepted for publication once it meets all outstanding technical requirements.

Kind regards,

Venkata Naga Srikanth Garikipati, PhD

Academic Editor

PLOS ONE
---

## [Editor Report · Acceptance letter]

29 Jun 2022

PONE-D-21-35995R1 

Associations of metabolic syndrome and metabolically unhealthy obesity with cancer mortality: The Japan Multi-Institutional Collaborative Cohort (J-MICC) Study 

Dear Dr. Arisawa:

I'm pleased to inform you that your manuscript has been deemed suitable for publication in PLOS ONE. Congratulations! Your manuscript is now with our production department. 

Kind regards, 

on behalf of

Dr. Venkata Naga Srikanth Garikipati 

Academic Editor

PLOS ONE